# Travelling with a Guide Dog: Experiences of People with Vision Impairment

**Jillian M. Rickly** [1,*] , **Nigel Halpern** [2] , **Marcus Hansen** [3] **and John Welsman** [4]

1 Nottingham University Business School, University of Nottingham, Nottingham NG8 1BB, UK
2 Department of Marketing, Kristiania University College, 0107 Oslo, Norway; nigel.halpern@kristiania.no
3 North Wales Business School, Wrexham Glyndwr University, Wrexham LL11 2AW, UK; marcus.hansen@glyndwr.ac.uk
4 Guide Dogs, Reading RG7 3YG, UK; john.welsman@guidedogs.org.uk
* Correspondence: jillian.rickly@nottingham.ac.uk; Tel.: +44-115-846-6493

**Abstract:** There is considerable research on people with vision impairment (PwVI) in the transport, travel and tourism sectors, which highlights the significance of real-time information and consistency in services to accessibility. Based on interviews with guide dog owners in the United Kingdom, this paper contributes an additional dimension to our understanding of transport accessibility for PwVI by focusing specifically on guide dog owners' experiences in the travel and tourism sector. A guide dog is more than a mobility tool, but a human–dog partnership that improves the quality of life for PwVI; however, it also introduces constraints related to the dog's welfare and safety. Further, lack of understanding of guide dog owners' rights to reasonable accommodation leads to discrimination through service refusals and challenges to service access. This paper concludes that the limited and inconsistent public knowledge of disability diversity has serious ramifications for transport accessibility and suggests specific industry and legislative interventions in response.

**Keywords:** vision impairment; guide dog; accessibility; disability; travel; transport

## 1. Introduction

Accessibility is contingent on the nexus of legislation and the built environment [1]. It is also essential to reducing inequalities, which is one of the United Nation's (UN's) Sustainable Development Goals [2]. In many countries, service providers have a legal responsibility to comply with disability legislation, such as the Disability Discrimination Act (1995) and the Equality Act (2010) in the United Kingdom (UK) [3], where data collection for this paper took place. Such legislation builds from the notion of reasonable accommodation as outlined in the UN's Convention on the Rights of Persons with Disabilities. Reasonable accommodation argues that it is discriminatory to refuse to make modifications or adjustments of "undue burden" in order to support the access needs of users. The UK's Disability Discrimination Act was expanded in 2005 with the specific intention of making public transport accessible for all.

However, there is growing critique that much of this legislation is outdated, as it focuses on broad categories of disabilities, while the wider and growing disability spectrum, such as hidden disabilities and sensitivities, are disregarded [4,5]. Specifically, these authors [6] argue that this legislation is particularly limited for people with vision impairment (PwVI), a theme to which this paper aims to contribute by examining the role of guide dogs in the access of PwVI to transport in the UK.

According to the NHS [7], there are about 2 million people living with sight loss in the UK, of which around 360,000 are registered as blind or partially sighted, with many charitable organisations dedicated to supporting PwVI through education, assistance services and training for assistive technologies. Guide Dogs for the Blind Association, known more commonly as Guide Dogs, is a UK-based charity that provides guide dogs

and other support services to PwVI. There are currently around 5000 working guide dogs registered in the UK [8].

The benefits of guide dogs to PwVI are well-established [9–14]. Most notably, guide dog owners experience increased independence, confidence and quality of life, as well as greater mobility in their local communities [14]. While others find that access to public spaces is improved with the acquisition of a guide dog, PwVI also periodically experience additional barriers when attempting to enter some pubs, hotels or public transport [11,15], suggesting a lack of understanding of guide dog owners' rights to reasonable accommodation. Further, some authors find that guide dogs introduce new responsibilities and stresses, even if the overall balance improves quality of life [16]. Thus, while a guide dog can assist in overcoming some barriers of vision impairment, it might also create additional constraints related to discrimination, as well as care and welfare concerns. As a result, this research contributes to our understanding of tiered constraints and barriers to travel for people with disabilities [17].

Despite the strong evidence that guide dogs improve mobility and independence in local communities [14], much less is known about the role of guide dogs in travel mobilities outside of one's everyday environment. As a result, this paper focuses on the experiences of PwVI travelling with their guide dog across different transport scenarios encountered during holiday travel. While there is a growing body of literature on the experiences of PwVI across the transport, travel and tourism sectors, it overlooks the guide dog as an active part of the experience and decision-making. Based on a qualitative study, which includes semi-structured interviews with 27 guide dog owners in the UK, this paper identifies several themes that highlight the barriers to transport access for PwVI travelling with a guide dog: service refusals, inadequate staff training, service access and dog welfare and safety.

The remainder of this paper is structured as follows: Section 2 provides a review of key literature on transport disadvantage and accessible tourism, travelling with vision impairment and travelling with a guide dog. Section 3 describes the qualitative approach taken in order to capture the experiential dimensions of travel with a guide dog. Section 4 provides a discussion of the main findings. Section 5 provides a conclusion that highlights the theoretical, practical and policy contributions, as well as recommendations for further research.

Additionally, we ask readers to note our usage of "people with vision impairment" (PwVI) for two reasons. First, while publications use different labels, such as visually impaired people (VIP), we support "people first" language that prefaces the individuality and dignity of people with disabilities [18]. Second, rather than "visual impairment", we use "vision impairment", as our work with sight loss charities and research participants revealed that among this community "visual impairment" has connotations suggesting impairment of a person's appearance, whereas "vision impairment" relates specifically to sight abilities.

## 2. Literature Review

Tourism mobilities encompass various modes of transport, with accessibility crucial across the journey. In the literature review that follows, we aim to relate transport disadvantage research specifically to the context of travel and tourism sector accessibility.

### 2.1. Transport Disadvantage and Accessible Tourism

Transport inequalities and social exclusion are more often situated within the concept of transport disadvantage. A review of this literature concludes that while progress has been made to address key issues, there is still much work needed to convince stakeholders of the value of social inclusion and researchers that more diversity of representation is needed [19]. Indeed, in the near decade following a call to action in this regard [19], we can observe more comprehensive coverage of disability and accessible transport in research, including studies of accessibility and barriers [20–24] and transport experiences [6,25]. Studies also look at strategies that might increase opportunities for mobility and reduce

exclusion, including assessing the effectiveness of policy interventions [26,27] or specific interventions such as the introduction of concessionary bus travel [28], as well as the role of household members in providing support to people with disabilities [29].

Within the travel and tourism literature, transport disadvantage is captured under the broader concept of accessible tourism. The impetus of this concept is the removal of barriers (physical, institutional, informational and attitudinal) across the industry, which encompasses a variety of transport and accommodation scenarios [30,31]. Accessible tourism is also attentive to the whole-of-life approach, which aims to be cognizant of the cross-generational market and the diversity of disability access needs [32]. However, barriers remain across the industry, in particular a general lack of training and awareness of disability diversity [17,33,34], which is illustrated by a pervasive "one-size-fits-all" approach to accessibility compliance [35]. Unfortunately, it is not uncommon for service providers to assume that by being wheelchair accessible, they are therefore accessible to all [36]. For example, PwVI are often unnecessarily brought a wheelchair in order to guide them through airports [35].

Accessible tourism research supports the social model of disability that aims to shed light on the disabling features of the social and physical environment [37,38]. Indeed, research shows that there is not a lack of desire to travel among people with disabilities (PwD), but rather greater constraints and barriers to negotiate in order to participate in travel [17,39,40]. This is what inspired one of latest conceptual efforts to capture the diversity of disability experiences in the travel and tourism sector [17]. By elaborating a tiered model of barriers and constraints, the authors illustrate: the constraints experienced by all travellers, the barriers faced by all PwD, the barriers faced by groups of people with shared disabilities (i.e., vision impairment, mobility impairment, cognitive impairment), and the individual's lived experiences of barriers related to their particular disability severity (and/or co-morbidities) [17].

Thus, this paper is a response to calls for greater diversity of representation of PwD's experiences in transport [19], as we focus on PwVI who are guide dog owners travelling outside of their everyday environments. Additionally, it supports the importance of a tiered model of constraints and barriers [17] by highlighting the multi-faceted experiences of travelling with vision impairment and with a guide dog, which both assists in overcoming some barriers while also presenting potential for further discrimination.

### 2.2. Travelling with Vision Impairment

PwVI are restricted from driving a car, thereby making public transport one of the few options available to them [6,41]. Yet, PwVI experience reduced confidence in using public transport [42,43]. There are many potential barriers to using public transport, including difficulties hearing audio announcements, navigating congested spaces, locating the correct bus or train car at a station, finding entrances/exits as well as seats, navigating transfer points, receiving notifications and real time information and obtaining assistance on site [6,44–49]. Some assert "the main need [for PwVI] is access to more and better *information*" [46] (emphasis in original, p. 457), and others more specifically identify the "fear of missing information," [50] (p. 152) as a major source of travel anxiety for PwVI. However, more recent research contends that consistency in design of transport, transit hubs and use of tactile paving is just as crucial to the ability of PwVI to create reliable mental maps, which support safe and confident movement throughout their journey [26,42,51,52].

PwVI use a variety of support and assistive technologies, such as screen readers to access textual information [53]. When it comes to wayfinding, GPS-based technologies are increasingly integrated into mobile applications and smart technologies to aid navigation [6,52,54–56]. However, these are most often used in conjunction with a guide dog or traditional assistive tools, such as long canes, as well as implementing the common practice of counting steps and memorising routes. Examining the experiences of PwVI using rail, some emphasise the importance of repetitive exposure and interaction with the same station to acquire and develop knowledge of the barriers in the form of a mental

map [42]. Advanced journey planning through digital resources and mobile applications are becoming essential for PwVI to gain a sense of familiarity and preparedness before beginning travel [6].

However, when travelling outside of their everyday environment, PwVI are more likely to rely on human assistance, in particular guiding [57,58]. This introduces further constraints for PwVI to access travel and transport. In the context of holiday travel, PwVI are much less likely to travel compared to people with other impairments [36,57]. In addition to the informational and physical barriers discussed above, holiday travel for PwVI is further constrained by attitudinal barriers [17] and negative experiences are more likely to become future demotivators to travel [42,59]. As a result, the availability of vision impairment services and staff support, as well as disability diversity staff training, more broadly, is essential [6,58,60].

Remarkably, while there is now considerable literature on transport, travel and tourism experiences of PwVI, one aspect missing from this body of literature is the guide dog [61]. The guide dog is scarcely mentioned, and when it is, it is more often considered tangential to the person's experience. However, as this paper demonstrates, the human-guide dog relationship is salient to understanding transport, travel and tourism experience, in particular as the dog has distinct welfare needs, but also opens up the potential for further discrimination along the journey.

*2.3. Travelling with a Guide Dog*

A guide dog is a domestic dog trained to assist PwVI to safely navigate past obstacles and hazards, such as objects, curbs and vehicles, once given a directional command [62]. There is a robust body of research detailing the benefits of guide dogs for PwVI, particularly in terms of increased independence, mobility, confidence, self-esteem, companionship and social interaction [9–14,16,63–65]. In fact, one study finds a preference for guide dogs among PwVI compared to other mobility aids, such as the long cane or technological alternatives [66]. For some, the presence of a guide dog enhances the "visibility" of the person's impairment, alerting others that they might require reasonable accommodation [16]. Other research has found that guide dogs brought an increase in positive social interactions for PwVI in both their personal lives and public settings [12,14].

Further, the close human–dog bond that develops as a result of their embodied working relationship is amongst the most noted reasons for their preference over other forms of assistance [13,62,63]. The dogs learn to understand their owner's cues and ways of communicating, while the person simultaneously learns the dog's personality and responsiveness. As such, it is an evolving relationship that is continually unfolding with time and experiences [13]. While technically a "working dog", guide dogs spend only a fraction of their time "at work" and the majority of their time they are in the role of "pet" or "companion" dog, thus further enhancing the bond [67].

Despite the many positive outcomes of guide dog ownership, living with and caring for a dog brings responsibilities and challenges [16]. Moreover, the loss of a guide dog can have severe emotional ramifications akin to the death of a family member or partner [12,68]. Additionally, the presence of a guide dog may present possibilities of discrimination and barriers to access [15,16]. Guide dog owners may struggle to locate appropriate facilities for their dogs and the role of the guide dog is also commonly misunderstood by the general public [4]. Indeed, PwVI have been refused access to public transport, restaurants, shops, hotels, among many others, when accompanied by their guide dog [4,69]. Importantly, while refusals are discriminatory, they can also have serious ramifications for the dogs' welfare and safety [70]. Therefore, the finding that inclusive attitudes towards guide dogs lead to more enjoyable tourism experiences for PwVI is not surprising [35].

This paper, thus, contributes to our understanding of the accessibility of transport for PwVI by highlighting specific barriers and constraints experienced when travelling with a guide dog. In so doing, it also calls attention to the implications of inadequate services for guide dog welfare and safety.

### 3. Methodology

This study uses a qualitative approach based on interviews with guide dog owners. Participants were gathered from the Guide Dogs' database of members who currently have a guide dog or have recently retired their dog and are on the waiting list for a new one. Using semi-structured interviews, questions were informed by the literature reviewed in Section 2 of this paper and were developed in collaboration with Guide Dogs' Research Team.

Qualitative approaches are considered useful and necessary in the context of research that focuses on the complexities of transport users' needs and experiences [6,58,60]. To gain a deeper understanding of guide dog owners' experiences, semi-structured interviews were employed to capture a more vivid picture of the significance and impacts of their experiences on their travel behaviour and decision-making [71,72].

After 27 interviews, we felt that saturation had been sufficiently reached so did not solicit further participants. Generally, saturation is achieved when enough data has been collected to allow for replication of the study and when any additional data does not facilitate further analysis [73]. The interviews lasted an average of 45 min. The semi-structured approach asked respondents to speak about themselves (i.e., vision impairment, work situation, dog ownership), their travel behaviour with a focus on travel outside their local area, and actual travel experiences with an emphasis on specific barriers and challenges associated with transport. Interviews were conducted by telephone and were audio-recorded with permission from respondents for transcription and analysis purposes. To protect the participants' identity, anonymous identifiers replaced their names and the names of their guide dogs. Following transcription of the recordings, transcripts were analysed with thematic analysis by two of the authors [74]. The thematic analysis process was two-fold. One author applied themes developed from a review of the relevant literature, specifically the themes of accessible tourism—independence, equity and dignity— were assessed. Then a second author reviewed the coding for consistency and applied an additional layer of codes identifying sub-themes. The themes and sub-themes are illustrated in Figure 1.

A summary of characteristics for the interviewees is shown in Table 1. Importantly, this research was conducted prior to the Covid-19 pandemic and, as a result, the impacts of the pandemic did not inform participants' responses.

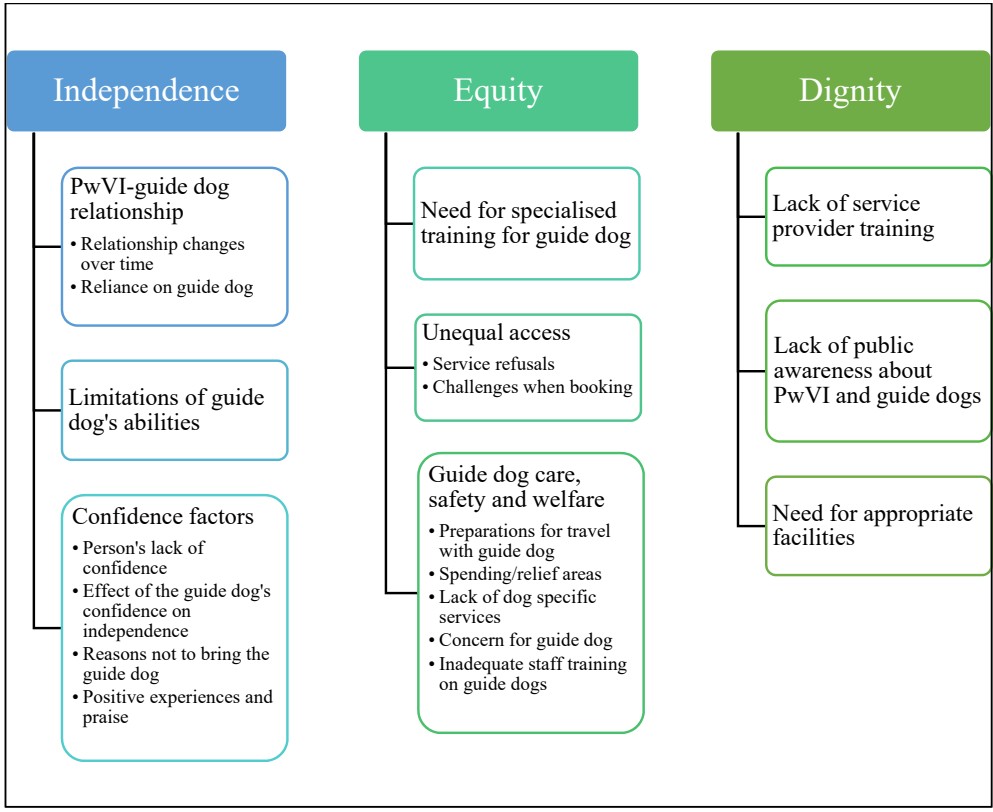

**Figure 1.** Thematic coding map.

**Table 1.** Interview participants.

| Participant | Gender | Age in Years (To the Nearest Decade) | Total Years with a Guide Dog |
|---|---|---|---|
| 1 | Male | 60 | 37 |
| 2 | Female | 30 | 17 |
| 3 | Female | 60 | 39 |
| 4 | Female | 60 | 22 |
| 5 | Female | 50 | 14 |
| 6 | Male | 50 | 6 |
| 7 | Male | 70 | 10 |
| 8 | Female | 50 | 7 |
| 9 | Female | 60 | 35 |
| 10 | Male | 20 | 10 |
| 11 | Male | 40 | 4 |
| 12 | Female | 60 | 21 |
| 13 | Female | 50 | 2 |
| 14 | Female | 30 | 14 |
| 15 | Female | 50 | 12 |
| 16 | Female | 40 | 25 |
| 17 | Female | 30 | 6 |
| 18 | Male | 30 | 3 |
| 19 | Male | 60 | 11 |
| 20 | Female | 60 | 42 |
| 21 | Male | 60 | 9 |
| 22 | Female | 50 | 18 |
| 23 | Male | 50 | 14 |
| 24 | Male | 60 | 21 |
| 25 | Male | 70 | 5 |
| 26 | Female | 40 | 10 |
| 27 | Female | 50 | 30 |

## 4. Findings and Discussion

In order to tease out the details and nuances of individual's travel experiences, we included broad questions on barriers and challenges, which allowed participants to tell their own stories and convey the significance of encounters. Analysis revealed several themes of importance to transport accessibility. In particular, we elaborate on service refusals, staff training, service access, and dog welfare and safety.

### 4.1. Service Refusals

Refusals of transport services featured prominently in the interviews. In particular, stories of refusals were exclusively related to taxis and private car hires. In fact, nearly every interviewee had experienced this on at least one occasion and the significance of these events had lasting impacts on their perceptions of access rights and overall stress and anxiety when travelling. Under part 12 of the UK Equality Act (2010) it is specifically against the law for taxis and minicabs to refuse service to a guide dog owner travelling with their guide dog, unless the driver has a medical certificate for exemption. As the following quotes demonstrate, participants have had their rights ignored in a number of ways, from refusal to accept a booking to last minute cancellations upon observing a guide dog in the company of the customer.

> "I phoned for a taxi and I said the normal spiel you know, and the woman said, 'Oh no, we can't give you a taxi, the driver won't take a guide dog'. And I said 'You are breaking the law', 'Well I don't care, the driver won't take a dog' [ . . . ] I really, really will do anything to avoid using a taxi."
>
> (Participant 9, guide dog owner of 35 years)

> "I used to use Uber before [ . . . ] when I got my guide dog, the minute they see the dog in 75% of the time it's a problem. [ . . . ] all of a sudden, I started having problems with Uber over cancelling. And what I found was the Uber driver would pull up, see I'd got a guide dog and just cancel and drive off. And of course, I wouldn't know they'd turned up."
>
> (Participant 13, guide dog owner of 2 years)

Such occurrences support research on greater difficulties to access when accompanied by a guide dog [4,57,69]. However, taxi refusals as a specific barrier to transport, mobility and independence has not yet been highlighted in the literature and indicates a novel finding of this research which is of considerable significance to the reasonable accommodation rights of PwVI.

Due to the prevalence of taxi and private hire car refusals, Guide Dogs have launched specific awareness campaigns and training programs. As a result, where these have been implemented, respondents reported improved access and very few challenges to using taxis.

> "In this area you would never get a taxi refusal [ . . . ] Because the borough investigates properly and would take action [ . . . ] For example they evoked all taxi drivers' licences and take them on an NVQ [National Vocational Qualification] course [ . . . ]. So all their drivers have had disability equality training."
>
> (Participant 22, guide dog owner of 18 years)

### 4.2. Staff Training

The knowledgeability of staff can be a key variable distinguishing between a positive and a negative experience for guide dog owners. The following quote from a participant travelling to Portugal demonstrates the significance of an inclusive and accessible air travel experience.

> "I can never have enough praise for Faro Airport [ . . . ] from the minute you arrive and the same coming back through when you're going home [ . . . ] They

go through step by step. They meet you off the plane. And then they literally took me right outside to Arrivals, to the person I was being picked up with."

(Participant 5, guide dog owner of 14 years)

Unfortunately, in the interviews, much more common were stories of inadequate staff training that included lack of knowledge about rights to reasonable accommodation, what documentation is required, and how to interact with PwVI and their guide dog. The following two quotes are from participants' experiences in airports, specifically.

"easyJet they did keep telling me I didn't have the right paperwork. But when I asked them what paperwork I needed they couldn't tell me."

(Participant 15, guide dog owner of 12 years)

"the biggest burden and the biggest negative to travelling is airport staff who are not properly trained. And because of that two things happen: one, you arrive at the desk and they say 'Oh, never seen a guide dog before, I'll just go and check with my supervisor' and [two] they disappear, leaving you standing there."

(Participant 7, guide dog owner for 10 years)

However, the following quotes, while based on train experiences, suggest broader misunderstandings about accessibility, by privileging wheelchair users at the expense of the guide dog's welfare and not considering the needs of the dog.

"They'll say 'oh no we can't reserve this seat for you and your guide dog because there might be a person coming on with a wheelchair who cannot get out of their wheelchair and we have to consider them first'. Well, I can understand that to a degree, however I also think, well hang on, my guide dog is a living thing, it doesn't need to be in the middle of the aisle where everyone's going to kick it."

(Participant 13, guide dog owner of 2 years)

"Usually on the train it can be a pain because when I've got special assistance. For example, they think that they're going to give me more room by putting me in a seat where there's a table in front of me. But the way the tables are, you've got two legs in front of the table, which means the dog can't fit underneath it. [ . . . ] You can't just spontaneously go for a trip on the train anymore, I mean not with the dog. [ . . . ] I want them to know that they shouldn't be touching the dog. It's all about the interaction."

(Participant 18, guide dog owner of 3 years)

Lack of awareness of disability diversity often presents in a "one-size-fits-all" approach to compliance [35]. While it is a positive indicator that in the first case the train staff were cognizant of wheelchair user needs, this came at the expense of the dog's welfare by putting it in a space of potential harm. Similarly, in the second case, while good intentions often lead to this participant being seated at a table, inadequate understandings of PwD, and guide dog owners specifically, mean that the arrangement actually causes further problems.

### 4.3. Service Access

A recurrent theme among the interview participants' accounts was the lack of consistency of available services and service access for guide dog owners. As argued in the literature, consistency is essential for PwVI to gain familiarity with transport services and build confidence as users [26,42,51,52]. However, the following two quotes show participants with exact opposite experiences of attempting to book flights with their guide dogs. While one found they could only complete the process by using an online form, the other was required to use a telephone in order to complete the booking.

"You can't just phone them [airline] up and say look, I'm on this flight and I've got an assistance dog [ . . . ] you've got to find specific kinds of forms on their website and fill in all the details and you've got to do it at least two weeks in

advance [ . . . ]. I don't know what you'd do if you didn't have internet access, it wouldn't be possible to travel I don't think."

(Participant 6, guide dog owner of 6 years in their 50s)

"The worst one is booking the guide dog because you know, you can go online, and you can book your travel arrangements for you and your partner [ . . . ]. Nobody's got an online booking process for a guide dog or an assistance dog. So, you're faced then with phoning them up and telling them. It takes at least half an hour extra."

(Participant 7, guide dog owner for 10 years in their 70s)

Further, because of the extra steps required of guide dog owners when making a booking, they may actually be excluded from accessing other services. As the following quote illustrates, the benefits of reasonable accommodation legislation create a new challenge in the booking system that does not allow the user to use advance check-in, as other passengers do.

"So, under reasonable adjustment they block out the seat next to you at zero cost. [ . . . ] Different airlines have different setups. And for a lot of them you end up with a separate reservation for the dog compared to you, which means that you cannot do online check-in in advance. [ . . . ] But where more and more people check-in online so many days in advance, most of the plane seating has already been given out. So that bit is still quite challenging, and I've known them actually have to stop people at the gate to move people and change people's tickets at the gate, which has delayed everyone."

(Participant 11, guide dog owner of 4 years)

### 4.4. Dog Welfare and Safety

Following on from the general lack of staff knowledge and training on PwVI with a guide dog outlined above, participants also noted specific shortcomings in terms of understanding dog welfare needs and services to care for them. Most specifically, when speaking about their guide dogs' welfare needs when travelling, participants noted the lack of toileting areas in transit spaces.

"Another thing that I'm conscious of is that most train stations don't have anywhere to spend your dog."

(Participant 17, guide dog owner of 6 years)

"It's pressing [ . . . ] for all airports to have toilets after security, so that you can take your dog to the toilet before it jumps on the plane."

(Participant 7, a guide dog owner for 10 years)

Relatedly, an experience by one participant in which airline staff asked if a guide dog could wear a nappy (or diaper), further illustrates the broad misunderstandings about guide dogs' training and abilities.

"The airline invents some rules that are out of line with reasonable standards. So, for example we were asked at Amsterdam Airport by KLM flying to London City Airport, if we had a diaper for the dog, a nappy."

(Participant 7, guide dog owner for 10 years)

Finally, in addition to toileting and care concerns, PwVI must also consider the safety of their dogs. While guide dogs can assist PwVI to navigate and avoid obstacles, in the case of the gap between the train and platform, the presence of a guide dog might amplify existing anxieties. For example, the following participant had a guide dog fall through this gap, illustrating the real danger to the health and safety of PwVI and to guide dogs when tactile paving is not consistent and attendants are not in place to assist travellers [6].

"I had an incident a long time ago where my first guide dog fell between the platform and the train. [ . . . ] the gap is too big and then they just don't make it. And because quite often it's slippery floors, they just slide off."

(Participant 15, guide dog owner of 12 years)

In fact, it has been observed that navigating this gap causes anxiety among rail users in general [75], so for PwVI who have a guide dog, they have additional concerns for themselves and their dog's safety.

## 5. Conclusions

This paper sits at a crucial intersection of transport disadvantage and accessible tourism research. Building upon broad understandings of the transport, travel and tourism experiences of PwVI [6,41–43,57,58,60], we focus specifically on those with a guide dog. This has several important implications for theory, practice and policy.

Theoretically, this paper adds a new dimension to our understanding of the travel behaviour and decision-making of PwVI, as it illustrates the barriers and constraints of living with vision impairment, but also those that come with being responsible for the care and welfare of a guide dog. Importantly, the human-guide dog relationship is more than a mobility tool, it is a partnership that develops over time enriching a companionship as they grow, trust and learn together [12,13]. While a guide dog improves the quality of life for PwVI and aids in overcoming many barriers to independence, it also presents new challenges. The guide dog is a living, breathing and feeling being whose welfare and safety demand consideration. This is why reasonable accommodation legislation extends to access for guide dogs in the company of PwVI. However, as this paper has demonstrated, the themes of service refusals and inadequate staff training were strong across our interview findings, suggesting the need for interventions in transport, travel and tourism sectors.

Experiences of refusals by PwVI were, overall, quite low across the various modes of transport used. However, nearly every interviewee had an experience of refusals related to taxis or hired cars. This points to the very real need for change by taxi and hired car services. In fact, Guide Dogs has made considerable effort in recent years to design awareness campaigns and training programmes to address disability diversity, working closely with local councils across the UK. As a result, our participants in these areas observed improvements in taxi services.

Relatedly, inadequate staff training on the rights of guide dogs, their welfare and safety needs, as well as how to interact with them while they are working were issues common across our findings. Despite a growing diversity of PwD representation in the academic literature on accessible tourism, transport disadvantage remains a challenge that the industry must address. There is a systematic lack of understanding of disability diversity, evidenced by the all too common "one-size-fits-all" approach to accessibility legislation compliance [35]. While it is a positive move in the right direction that tourism transport is increasingly aware of wheelchair accessibility needs, there remains much work to be done on accessibility across the disability spectrum and towards inclusivity of those living with multiple disabilities. Specifically, this paper illustrates the complexities of accommodating multiple accessibility needs, in this case for vision impairment and for guide dogs.

It is this intersectionality of multiple access needs that most warrants updating disability legislation. Currently, many categories of disability are included in disability legislation, such as vision, hearing, mobility, cognitive, emotional/psychological and hidden; however, there is less consideration given to the fact that many people live with multiple disabilities. In fact, our findings related to service access illustrates good efforts at accommodating PwVI who have a guide dog. Yet, an unintended outcome has been inconsistency of those services and unequal access to additional services afforded to other passengers, such as advance check-in for airlines.

This research took place prior to Covid-19; nevertheless, it highlights the importance of inclusive accessibility as the transport, travel and tourism industries recover and rebuild

for the future. Indeed, many businesses are viewing the pandemic as an "opportunity" in which to institute more sustainable practices. However, Covid-19 has also shed light on the most vulnerable of our society and the challenges they face in equal access to transport services. In particular, it revealed just how far we have to go to reach many of our Sustainable Development Goals, in particular Goal 11 Reduced Inequalities. Thus, future research should broaden the scope of this paper's focus on guide dogs, by looking at all assistance dog users. Assistance dogs are employed to overcome many impairments that challenge independent lives. Examining this travel segment will provide further insight into the diversity of disability and where accessibility must extend its reach.

**Author Contributions:** J.M.R.—40% (Conceptualisation; Methodology; Formal analysis; Writing— Original draft, review and editing); N.H.—30% (Conceptualisation; Methodology; Formal analysis; Writing—Original draft, review and editing); M.H.—20% (Conceptualisation; Methodology; Formal analysis; Writing—Original draft, review and editing). J.W.—10% (Methodology; Manuscript review; Validation). All authors have read and agreed to the published version of the manuscript.

**Funding:** This work was supported by the ESRC Small Steps, Small Business Fund and the ESRC Business Boost Small Grants and Follow-on Funds.

**Institutional Review Board Statement:** The study was conducted according to the guidelines of the Declaration of Helsinki, and approved by the Research Ethics Committee of Nottingham University Business School (Project "Assistance dogs on holiday" on 8 August 2018).

**Informed Consent Statement:** Informed consent was obtained from all subjects involved in the study.

**Data Availability Statement:** Due to contractual agreement with Guide Dogs, data are not publicly available. However, a full report of the project can be found at the following open access repository: https://rdmc.nottingham.ac.uk/handle/internal/8298 accessed on 19 November 2020.

**Acknowledgments:** The authors wish to extend their appreciation to Guide Dogs for their collaboration in this research. This manuscript was reviewed and approved by Guide Dogs Research Team on the 8 January 2021, as per contractual agreement, prior to submission to this journal.

**Conflicts of Interest:** One of the authors, John Welsman, works for Guide Dogs, which is disclosed here a potential conflict of interest.

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
