# Peer review of "Travelling with a Guide Dog: Experiences of People with Vision Impairment"

_sustainability, doi:10.3390/su13052840_

Round 1

Reviewer 1 Report

An excellent paper and critically needed. Well supported by the lit review. Methods straight forward and precise. Results and conclusions clearly articulated and organized. Well done 

Author Response

An excellent paper and critically needed. Well supported by the lit review. Methods straight forward and precise. Results and conclusions clearly articulated and organized. Well done 

Thank you very much for your support.

Reviewer 2 Report

I think the work is well written and certainly deserves to be published especially for the important topics covered. I have only a few small suggestions for the authors to improve the immediate understandability of the results (please see below).

A figure (or more) representing the results is missing and must be added to improve the immediate reading of the data.

For example, I suggest creating histograms (or families of histograms) each of which represents a “themes of particular importance to transport accessibility”and translating the answers given by the interviews into terms of percentage for the presence/absence of problems or inconveniences.

Even a table with the questions asked would certainly improve understanding of the text.

Line 229:” After 27 interviews we felt that saturation had been sufficiently reached so did not solicit further participants”.

It is not clear what is the principle behind this saturation: is there a statistical test? Please give reasons for this choice.

An update on the study of the perceptual physiology of dogs (especially the vision) should also be mentioned in the text  “section 4.4 Dog welfare and safety” and considered in the future for the realization of any environmental facilitations that help the mobility and orientability of the dogs themselves. (The following paper should be cited and discussed here:” Siniscalchi Marcello, d'Ingeo Serenella, Fornelli Serena and Quaranta Angelo 2017Are dogs red–green colour blind?R. Soc. open sci.4170869).

Line 399:“Importantly, the human-guide dog relationship is more than a mobility tool, it is a partnership that develops over time enriching a companionship as they grow, trust and learn together [12,13]”.

This paragraph is extremely important, in fact it has been shown that the human-dog relationship is a real attachment bond that was previously believed to exist only in primates (please see and cite the following paper:” Siniscalchi M, Stipo C, Quaranta A (2013) "Like Owner, Like Dog": Correlation between the Owner's Attachment Profile and the Owner-Dog Bond. PLoS ONE 8(10): e78455. https://doi.org/10.1371/journal.pone.0078455).

Author Response

I think the work is well written and certainly deserves to be published especially for the important topics covered. I have only a few small suggestions for the authors to improve the immediate understandability of the results (please see below).

Thank you for your support. We have attended to the suggestions made, with further details below.

A figure (or more) representing the results is missing and must be added to improve the immediate reading of the data.

For example, I suggest creating histograms (or families of histograms) each of which represents a “themes of particular importance to transport accessibility” and translating the answers given by the interviews into terms of percentage for the presence/absence of problems or inconveniences.

Even a table with the questions asked would certainly improve understanding of the text.

We appreciate the reviewer’s suggestions. Importantly, though, this is a qualitative study, which is why we have not included a quantitative element to the analysis. However, in light of the reviewer’s suggestions, we have added a Thematic Coding Map and text about the thematic analysis of the interviews (Figure 1; Lines 266-273). In addition, Lines 252-256 describe the semi-structured line of questioning that was used for the interviews.

Line 229:” After 27 interviews we felt that saturation had been sufficiently reached so did not solicit further participants”.

It is not clear what is the principle behind this saturation: is there a statistical test? Please give reasons for this choice.

Saturation is a methodological concept used in qualitative research. We have added further explanation of the concept and citation [73] to justify this approach. Please see Lines 250-252.

An update on the study of the perceptual physiology of dogs (especially the vision) should also be mentioned in the text  “section 4.4 Dog welfare and safety” and considered in the future for the realization of any environmental facilitations that help the mobility and orientability of the dogs themselves. (The following paper should be cited and discussed here:” Siniscalchi Marcello, d'Ingeo Serenella, Fornelli Serena and Quaranta Angelo 2017Are dogs red–green colour blind? R. Soc. open sci.4170869).

Thank you for sharing this paper on dog vision. We agree that an understanding of dog’s vision would be useful for guide dog training, but that is not the focus of this paper. Rather, the focus of this paper is on transport policy and overcoming transport inequalities for people with vision impairment.

Line 399: “Importantly, the human-guide dog relationship is more than a mobility tool, it is a partnership that develops over time enriching a companionship as they grow, trust and learn together [12,13]”.

This paragraph is extremely important, in fact it has been shown that the human-dog relationship is a real attachment bond that was previously believed to exist only in primates (please see and cite the following paper:” Siniscalchi M, Stipo C, Quaranta A (2013) "Like Owner, Like Dog": Correlation between the Owner's Attachment Profile and the Owner-Dog Bond. PLoS ONE 8(10): e78455. https://doi.org/10.1371/journal.pone.0078455).

We agree with the reviewer’s sentiment that “the human-dog relationship is a real attachment bond”, and that this has been well established in the literature more broadly. However, this paper is specifically about the human-guide dog relationship, which is why we did not originally reference this literature. Nevertheless, we have incorporated the recommended paper, but in a different location to the reviewer’s suggestion. In the suggested location we are specifically speaking to the human-guide dog relationship. Instead, we have added the reference to Line 219 where there is discussion of human-dog bonding more generally.

Reviewer 3 Report

I have no suggestions for improvement but simply want to thank the authors for an important and well executed addition to the literature on guide dogs' potentials and barriers.

Author Response

Thank you very much for your support.